# Using Social Media Camping Data for Evaluating, Quantifying, and Understanding Recreational Ecosystem Services in Post-COVID-19 Megacities: A Case Study from Beijing

**Haiyun Xu [1], Guohan Zhao [2], Yan Liu [3] and Meng Miao [4],***

1   College of Architecture and Urban Planning, Beijing University of Civil Engineering and Architecture, Beijing 100044, China; xuhaiyun@bucea.edu.cn
2   Research Center for Built Environment, Energy, Water and Climate, VIA University College, Banegårdsgades 2, 8700 Horsens, Denmark; gzha@via.dk
3   Department of Geoscience, Southwest University, Chongqing 400715, China; bluefire612@163.com
4   College of Finance, Renming University of China, Beijing 100872, China
*   Correspondence: miaomeng@ruc.edu.cn

**Abstract:** Recreational ecosystem services (RESs) are the diverse recreational opportunities provided by nature to humans, which contribute to the improvement of public health and social well-being. The use of online social media is an efficient method for quantifying public perceptions of recreational ecosystem services (RESs) delivered by a given landscape. With the continuously changing demand for nature-focused outdoor recreational activities since COVID-19, camping has become the fastest-growing outdoor leisure activity in megacities and a key indicator for how people perceive and value the RESs provided by the landscape. Such unexpected changings triggered by COVID-19 have further led to an imbalance between demand and supply, which results in fierce conflicts in urban green space management. This study presents a spatial pattern analysis of how people perceive RESs in a megacity-scale case study of Beijing using geo-tagged camping notes posted on Little Red Book (LRB). We employed these camping notes in the context of a megacity to (i) map public camping behaviors patterns in urban green spaces, (ii) evaluate spatial clusters of high/low RESs, and (iii) investigate the relationship between RESs, local landscape features, and gender through correspondence analysis. Our results show that considerable spatial clustering of camping behaviors was observed in both suburban and urban green spaces. However, suburbs revealed a substantially higher RES value than central urban areas. In addition, water bodies were discovered to have remarkably low RES, while grassland and urban forests were found to have a close link with higher RES. In addition, significant gender preferences have been discovered, where female visitors prefer to camp in grassland, and male visitors favor bare ground and urbanized regions. Our findings would assist decision-makers in optimizing urban green space planning and management, adapting to fast-changing public camping demands in the context of the post-COVID-19 era. Findings also contribute to the literature by applying spatial analysis of social media data to understand public outdoor recreation activities and perceived value for megacities' green space management.

**Keywords:** recreational ecosystem service; camping; outdoor recreation; social media; urban green spaces planning; post-COVID-19; public well-belling

## 1. Introduction

Cultural Ecosystem Services (CESs) pertain to the notion of ecosystem services, which are considered the primary quantification tool for valuing the human benefits gained from nature [1,2]. CESs incorporate the complete range of non-material benefits related to human perceptions of natural landscapes, such as aesthetics and a sense of place. Among them, Recreational Ecosystem Services (RESs) can provide people with increased physical health (e.g., through exercise) and psychological and emotional well-being [3]. Hence, it



is regarded as a specific recreational metric that quantifies the individual benefits gained from nature. RES spatial assessments have been explored from a variety of perspectives and sources during the past decades. For expert-based techniques employing theoretically-derived spatial indicators [3], a landscape quality index has been suggested to map the backdrop of outdoor leisure activities at various spatial scales [3]. In contrast to systems relying purely on the opinions of experts, a public-participation approach can generate a more precise spatial pattern of RESs provided by users. Consequently, geo-located data are often regarded as a more efficient source for assessing and analyzing recreational visiting rates [4,5], CES values, and motivations [6]. Here, some scholars [7] conducted a public participation GIS using both small-group workshops and larger-scale home sampling to obtain recreational information from the public for the design of Chugach National Forest. In addition, some scholars [8] evaluated the recreational values for cultural landscape corridor planning in China using PPGIS and local resident interviews and surveys.

Beyond traditional data sources, social media has attracted more attention than traditional data sources due to its crowdsourcing data and often updated data properties [9,10]. Indeed, numerous social media platforms provide geo-tagged information submission and sharing and then archive millions of user contributions into big data hubs [10], which enables a more efficient data collection technique than traditional human-to-human survey methods. In data-sparse environments, some authors [11] have discovered and characterized the areas of high recreational value in data-sparse environments by assessing low or infrequent visitor metrics represented in social media data. In addition, these crowdsourcing data have been updated in real-time, which means that new submission data incorporating real-world dynamics are constantly assimilated into available databases, thereby providing a more accurate and timely data-driven basis to reflect real-world dynamics than traditional data source approaches. Here, some scholars [11,12] have used social media data for RES evaluation to capture the temporal dynamics of recreation values, thereby shedding light on seasonal changes in RES with a spatial dimension.

With the demonstrated crowdsourcing and timely-updated features, we would further explore its potential on large-scale cases such as megacities, thereby providing a more comprehensive view and time-efficient solution to comprehending the ever-changing public perception and updated demands for a given urban landscape than traditional approaches. Since the COVID-19 pandemic, outdoor recreation activities have been severely restricted in the majority of megacities [13,14]. Meanwhile, such a global pandemic has had a profound effect on people's daily lives, extending its influence on the public perception and needs for urban green space. The substantially accumulated demands on natural experience have promoted camping transforming into the most popular outdoor recreational lifestyle. Nowadays, camping has also been shifted into a primary metric for RES assessments [15,16]. So far, little research has focused on the newly emergent "supply-and-demand" imbalance brought about by COVID-19. Consequently, it is important to collect data and evaluate how visitors perceive the RES value supplied by various urban green spaces in megacities in order to better meet public demands in the post-COVID-19 future. We believe that social media data can provide a practical answer for identifying and comprehending such newly developed supply and demand imbalances.

The primary objective of this study is to identify the public's perception of the recreation value of green spaces in the case of a megacity, Beijing. This is accomplished through the use of geo-tagged social media data of individuals' camping notes. Here, we define camping as outdoor recreational activities involving daytime or overnight stays in natural environments, which take occur in both urban and rural settings and provide people with a variety of recreational benefits. Camping in rural areas typically entails outdoor pursuits such as hiking, canoeing, climbing, fishing, swimming, kayaking, shooting, stone skipping, and bird watching. Urban camping activities include picnics, bicycling, Frisbee, dog walking, and football [17–19]. Thus, camping covers a variety of human recreations in natural environments and might be used as a major indicator for determining the RES given by green space. Using geo-tagged camping notes posted on Little Red Book (LRB),

we performed a spatial pattern assessment of how people perceive RES in a megacity-scale case study of Beijing. We outline the research objective following three steps: (1) To map patterns of public camping behavior in verdant urban spaces. We can comprehend the general geo-differences of public camping behaviors and identify popular camping areas based on the spatial distribution pattern of public camping behaviors. Based on this outcome, planners and decision-makers will be able to identify the public's camping requirements and formulate appropriate policies to manage the overall supply and demand of green space and public outdoor recreation. (2) Evaluate the spatial concentrations of high/low RES. Through the spatial appraisal of public-perceived RES via social network data, we could investigate the high or low RES value they gained in each camping site and comprehend the geographic distribution of significant camping sites providing high or low RES value across the entire camping area. Planners could make targeted investments and differentiated enhancement policies in these sites with high or low values, which could have implications for the future management of green space. (3) Examine the relationship between RESs, local land cover characteristics, and gender. As previous CES studies have demonstrated that public perceptions of landscape value may be affected by local land cover characteristics [1,8], we investigate this relationship in RES value to determine which land cover characteristics in camping sites are associated with public preferences and perceptions of high or low RES value there. On the basis of this result, future planners will be able to adjust and manage various land cover characteristics in green space management policies in order to better satisfy the preferences of the public of different genders and enhance their camping experiences for the purpose of obtaining recreational value. Therefore, our study finally raises the following research questions:

(1) What is the spatial distribution of camping in Beijing?

(2) How does the public perceive the benefit of RES in these camping areas?

(3) What is the relationship between RES value and land cover characteristics and the gender of visitors?

Our findings would contribute to an updated understanding of the public's perception of RES provided by Beijing's green spaces in the camping trend influenced by the COVID-19 pandemic. The answers to these three primary queries may also contribute to the optimization of future green space planning and recreation policy management in response to changing public outdoor recreation demands.

## 2. Materials and Methods

### 2.1. Study Area and Camping Trend in Post-COVID-19

We selected Beijing, which has a total administrative area of 16,410.5 km$^2$, as the case study. It consists of two principal parts, comprising the central urban area and the suburbs (See Figure 1a). The central urban area (official Chinese name: Chengliuqu, means six distrscts in the central urban area) encompasses Beijing's ancient town, the urban environment with the highest population density [20]. The central urban area is comprised of six districts: Dongcheng, Xicheng, Haidian, Chaoyang, Fengtai, and Shijingshan (See Figure 1a in red). According to the population data from the Government of Beijing Municipality, we illustrated the distribution of population in each district in Figure 1b. Chaoyang district has the highest population ($n$ = 3,449,000), followed by Haidian district ($n$ = 3,130,000) (See Figure 1b). The historic old town is situated in the heart of the central urban area and is comprised of Dongcheng and Xicheng, which have respective populations of 110,000,000 and 700,000,000 (See Figure 1b). Beijing's suburbs consist of ten districts: Shunyi, Tongzhou, Daying, Fangshan, Mentougou, Changping, Pinggu, Miyun, Huairou, and Yanqing (Figure 1a, white text). The total population of the suburbs is 10,910,400 (Figure 1b). Figure 1b depicts that Changping has the greatest population ($n$ = 2,270,000), followed by Daxing ($n$ = 1,993,600) [20] (See Figure 1b). Generally, suburban areas are predominantly comprised of freshly developed metropolitan areas and abundant natural resources.

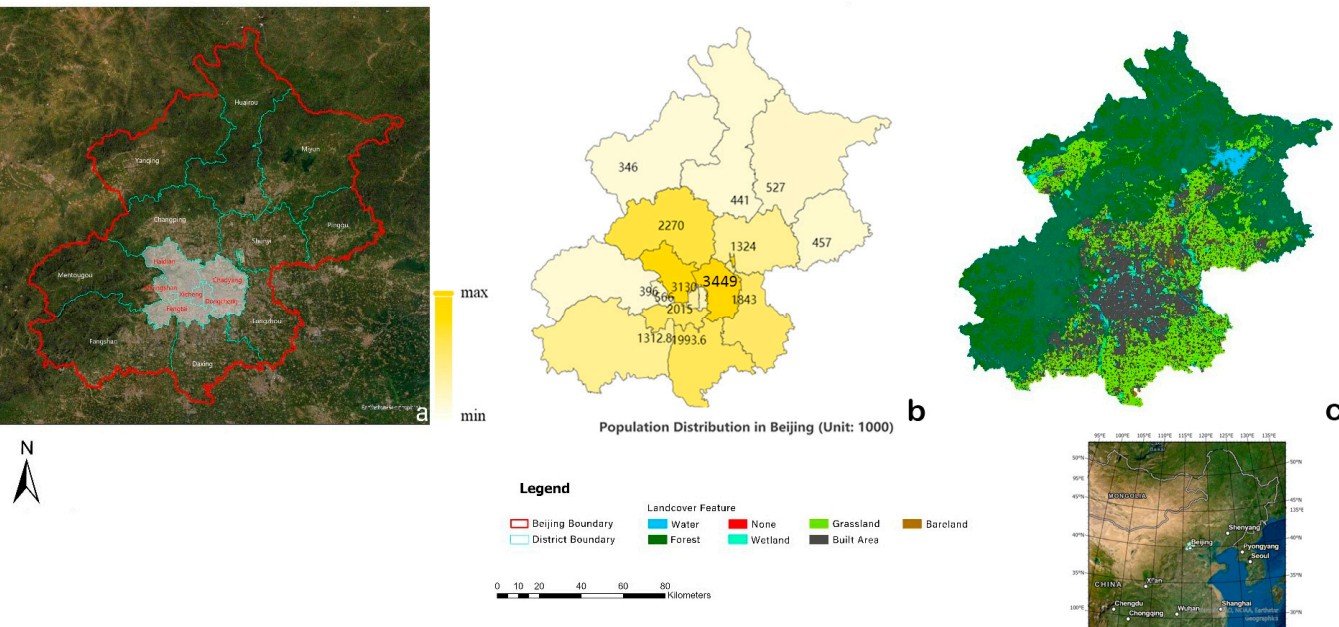

**Figure 1.** (**a**) the location and boundary of Beijing and its district divisions, where the grey regions with the central red fonts represent the central urban areas and the rest regions within the Beijing boundary represent the suburban areas. (**b**) The distribution of population in all districts. (**c**) the various green spaces in Beijing.

Beijing is the capital of China, a political, economic, and cultural hub, and the location of the royal historical garden green parks. Beijing has various green spaces in the central urban area and the suburban area (Figure 1c), such as urban parks, scenic areas, habitat conservation areas, forest parks, wetland parks, country parks, and diverse rural landscapes in the east and north mountain areas and plain area in the south, with land cover features such as forests, waterbodies, wetland, bare land, and grassland. These green places offered numerous CES, including recreational possibilities and RES values.

Since 2020, the pandemic has impacted public tourism activities and outdoor behaviors. In China, particularly after the COVID-19 outbreak, outward travel and outdoor recreational activities have been restricted, with safety being the top priority for outdoor activities [13,14]. Camping has become the most popular and fastest-growing outdoor lifestyle, especially for megacities in the post-COVID-19 age, particularly in Asia [15], as a result of a public need for a natural experience that has accumulated substantially throughout lengthy years of outdoor activity suppression. After that, camping flourished and gained significant public attention. As such, 2020 is called "the first year of Chinese camping" in social media by the tourism industry academy. In this trend, camping is becoming the most popular option for people to explore nature and gain leisure opportunities from the landscape in China in the post-COVID-19 era. According to Chinese social media data, search engine usage has surged by a factor of four. In 2021, the Chinese camping market is expected to rise by 75 percent and reach 29.90 billion yuan (4.69 billion US dollars) in value [17]. Beijing is one of the megacities with the fastest-growing camping tourism industry [17]. At the same time, despite the impact of the pandemic lockdown on construction projects in Beijing, the Beijing Municipal of Forestry and Park Bureau has promoted urban park enhancement projects since 2021. It included some micro-community green space initiatives, 26 new urban parks and 4 forest parks [21]. As one of the primary objectives of park construction projects in Beijing is to increase the green cover and park area per capita in this megacity, these park projects have a tendency to emphasize their ecological function. For micro-community green space improvement initiatives, decision-makers primarily focused on enhancing facilities for all ages, such as playgrounds for children and sports facilities for the elderly. However, camping recreation was rarely considered in the

aforementioned new urban park and green space projects. In addition, few parks have been implemented for opening, constructing, and designing camping-specific areas, despite the fact that it is the fastest-growing outdoor activity in the post-COVID-19 era. Moreover, the lagging nature of existing park regulations has left camping in Beijing's urban park management in a grey area, as it is a relatively new form of public recreation and use of green space. Camping pursuits may be evicted, acquiesced, or permitted in urban parks and green spaces based on the discretion of park and site administrators. Consequently, it is time to address this emergent public demand for the use of green space and to optimize future green space planning and recreation policy management in response to the shifting outdoor recreation public needs of the post-COVID-19 era.

### 2.2. Data Collection and Preprocessing

We utilized LittleRedBook (LRB) social media data to acquire citizen-uploaded data from 2020 to 2022. LRB was founded in 2013 as a social media site for sharing photographs of personal lifestyle achievements. It has been referred to as "China's Instagram". In 2019, LRB had more than 300 million registered users and more than 85 million monthly active users [22]. People can contribute notes and photographs with names and descriptions, share them, and add information to them on the website. Once members have uploaded their notes to LRB, the notes will include geo-location data. In this work, we extracted the original note data using the Application Programming Interface (API) provided by LRB. The terms forest, park, water, wetland, river, lake, and grass were used in conjunction with the phrase "camping" to filter the database of the social network to collect only relevant notes. The selection of keywords was predicated on the premise that they pertain to natural resources with potential for RESs [18,19]. We collected all camping notes using a Python code that systematically mined the geographic coordinates included with the posts on LRB. Finally, we identified 3680 geo-tagged sites containing camping-related notes from 2020 to April 2022. Each of these camping notes is then shared together with the name of the point of interest (POI), its latitude and longitude, images such as photos and hiking maps, topics, detailed text contents of their camping experiences, and the number of "likes" it has received. The latitude and longitude data indicated the precise geo-referenced location where this letter was uploaded online in real-time. POI reveals the user's camping location in accordance with note-described occurrences. Thus, we defined the social media data we used in this study as geo-tagged camping notes with photos and text presenting public camping experiences. However, we discovered that not all recorded note locations definitely belonged to camping sites or green spaces, as some people may upload their notes from their homes. To address this issue, we manually reviewed the data that had been collected by looking at all POI names and removing notes whose real locations did not correspond with campgrounds or green spaces, such as notes posted in shopping malls and Central Business District (CBD) buildings. Moreover, considering the impact of the pictures themselves in camping notes on the public's perception and preferences, we deleted the notes containing images focusing on people (such as the user's own portrait during camping events) and kept only the images of the scenery and camping-related activities, to ensure that the public perceives the same type of camping notes as much as possible in an effort to reduce the interference caused by the people's favorite pictures. As a result, our final tally was reduced to 2971 notes following this data cleansing process. To see the geo-locations of the data over the entirety of Beijing, the data gathered from LittleRedbook (LRB) was converted into GIS point data based on each post's geo-tagged location (i.e., latitude and longitude) using the WGS 1982 projection.

### 2.3. Data Analysis

To understand the GIS-point data, the general data analysis was conducted in terms of (i) spatial pattern analysis, (ii) hot/cold point cluster analysis, and (iii) land-cover correspondence analysis.

### 2.3.1. Kernel Density Analysis

Kernel density analysis is an ArcGIS tool for spatial analysis that calculates the density of features in a neighborhood surrounding those features. It calculates a magnitude-per-unit area from point or polyline features by fitting a smoothly tapered surface to each point or polyline using a kernel function [18,23]. The kernel density analysis was used to determine the spatial cluster of GIS points in order to better understand the spatial camping pattern. Based on the point density of the scattered GIS points, the kernel density analysis generates a smoothly curved circular surface raster, where each cell value indicates the expected number of points [23]. This was accomplished in two steps. To begin, we gather the occurrences of camping events within a 1000 m search radius. Second, we use a decaying factor of 2 to extend their aggregate value into neighboring raster grid cells. As a result, such a technique makes an explicit depiction of the geographical clustering of our mapped GIS points possible. In this study, we calculated kernel density estimation with an output cell size of 10 m; thus each cell reflected the expected point (note) count in 10,000 $m^2$ cells.

### 2.3.2. Cluster/Outlier Analysis for Hot/Cold Point Classification

Cluster/outlier analysis (Anselin Local Moran's I) is a spatial analysis tool in ArcGIS that uses the Anselin Local Moran's I statistic to identify statistically significant hot spots, cold spots, and spatial outliers [24]. The Anselin Local Moran's I is a common statistical tool to identify local cluster and spatial auto-correlation in given datasets [24].In this case, the Cluster/outlier analysis based on Anselin local Moran's I allowed us to distinguish between high/low RES value provided by green spaces (i.e., hot/cold points) among transformed GIS points mapped by public geo-tagged camping notes. The values assigned to each geo-tagged camping note are determined by the public's perceived RES here. Each camping note that users submit has the potential to reflect the human-perceived RES value of established urban green spaces. Here, we chose the number of "likes" on each note as the metric to quantify the RES values corresponding to each geo-location. A "like" symbolizes the reception of these messages and their apparent interest in the uploaded information, according to the social media study [25–27]. It was feasible to measure user engagement and investigate individual nature tourism experiences by analyzing the "like" statistics for these posts. In this case, "like" data can convey people's perceptions and interests in their nature tourism experience. Thus, from a public standpoint, the number of "likes" can indicate the RES value of local landscapes.

Based hereon, the method produces a map that highlights local areas with high or low Anselin local Moran's I values ranging from −1 to 1. A high positive value indicates positive spatial autocorrelation, where the target value is identified as similar to its neighborhood, and thus their corresponding locations are considered spatial clusters, including high–high clusters or low–low clusters. Instead, a high negative local Moran's I value implies a potential spatial outlier differing from the value identified from its surroundings, including high–low outliers and low–high outliers. In this case, other than spatial outliers, we are particularly interested in spatial clusters, and thus points with positive values were chosen. In addition, to ensure a sufficient degree of significance, we implemented a *p*-value less than 0.05, where a 95% confidence interval was used to reject the null hypothesis that the observed pattern was created by a random process. Therefore, hot points in this context defined the regions with statistically significant high RES values surrounding a high RES value neighborhood. In contrast, cold points were statistically significant low RES values surrounding a low RES value neighborhood.

Furthermore, based on the locations of the clustering analysis's hot and cold points, we randomly followed and invited three users who posted camping notes in each hot and cold point location for online interviews. Typically, two or three out of three individuals responded. For those who responded to our online semi-structured interview surveys, we first confirmed that they had visited the location themselves. The interviews then centered on the query, "How is your camping experience there?" In addition to their camping notes,

respondents were asked to describe their camping experiences at each location. We finally documented the text of these experiences and quoted some of them in subsequent sections.

2.3.3. Correspondence Analysis

We conducted correspondence analysis based on the identified hot/cold points to study further why hot/cold points were generated, taking into account the supplied land covers and the visitor's gender.

Correspondence analysis is a multivariate statistical tool developed by Jean-Paul Benzécri in 1973 to visualize the relationship between categories on a 2D map [28,29]. Differing from principal component analysis, it is applicable to categorical data as opposed to continuous data. Its purpose is to display any structure concealed by the multivariate data table setting in a biplot. Considered highly correlated are any two points, variables, or characteristics that are close together on the biplot. This analysis reveals the relative relationships between and within different groups of variables and is utilized in a variety of fields, such as medical research and marketing research [30]. In landscape management research, some authors have utilized correlation analysis to demonstrate the relationship between the public's perception of the CES and their diverse social backgrounds, such as age and education, in East Germany [1]. Some authors have investigated the relationships between public well-being, the socio-cultural characteristics of residents, and the perception of ecosystem service benefits in 13 European sites using correspondence analysis [31]. For landscape corridor planning, previous scholars conducted a correspondence analysis to examine how public perceptions of CES relate to local land-cover characteristics and public attitudes toward developing these lands [8]. On the basis of past practices, we intended to investigate the relationship between the public's perception of RES during their camping events, the land cover for their camping events, and their social characteristics. In this case, we postulate that different land-cover types may reflect on different key landscape elements in green space, which are linked to visitor decisions and experiences. Furthermore, gender differences in attitudes about camping activities may influence how the same urban green area is seen by men and women. So, based on the spatial relationship between hot/cold points and the land cover raster, as well as the spots in different districts, we created a cross-table that elaborates on the frequency of occurrence of hot/cold spots for various land cover types. The high-resolution land cover data was taken in 10-m resolution from Esri 2020 Land Cover (https://livingatlas.arcgis.com/landcover/, accessed on 1 June 2023). To ensure excellent data quality, the dataset was interpreted from Sentinel-2 Earth global observation photography, and the classifier was trained with billions of datasets from human-labeled image pixels. The land-cover dataset covers the entire city of Beijing and includes six major types of landscape features: water bodies, forests, built-up regions, grasslands, and bare land. Meanwhile, the original anonymous datasets were used to produce gender classifications for each note. Following the import of this data into Python's statistic correspondence analysis package, a correspondence analysis plot was generated, which shows the relative distance or closeness of the relationship between different points in the plot, reflecting lesser or greater degrees of correlation between the compared objects.

## 3. Results

### 3.1. The Spatial Pattern of Camping Behaviors

Figure 2a depicts the map of hotspots derived from kernel density analysis. In general, more camping behaviors (*n* = 2209) were seen in the suburbs than in the urban core. The Pinggu district drew the greatest number of campers (*n* = 494) of the regions surveyed (*n* = 452) (Figure 3b). According to the geo-referenced locations of 2971 camping notes gathered, six spatial camping clusters with the darkest color (in orange and red) were identified around Beijing. Table 1 lists the descriptive statistics for the areas with the highest population density. Similar to the spatial distribution, only one spatial grouping was found to be intersected within the central urban area, while the remaining seven spatial clusterings

were located in suburban areas. The core area clustering is predominantly located to the northeast of the city's central limit. We compared the 2971 locations of camping notes obtained with the Beijing POI names to obtain the total number of camping notes within each clustering area. According to the names of the POIs, this hotspot was comprised of urban parks, such as Chaoyang Park, whose landscape characteristics included an artificial lake, forest, lawn, and farmland. In contrast, the most densely populated suburban neighborhood is the Jinhaihu area, which is renowned for its expansive lake, mountains with colorful forests, and rolling lawns.

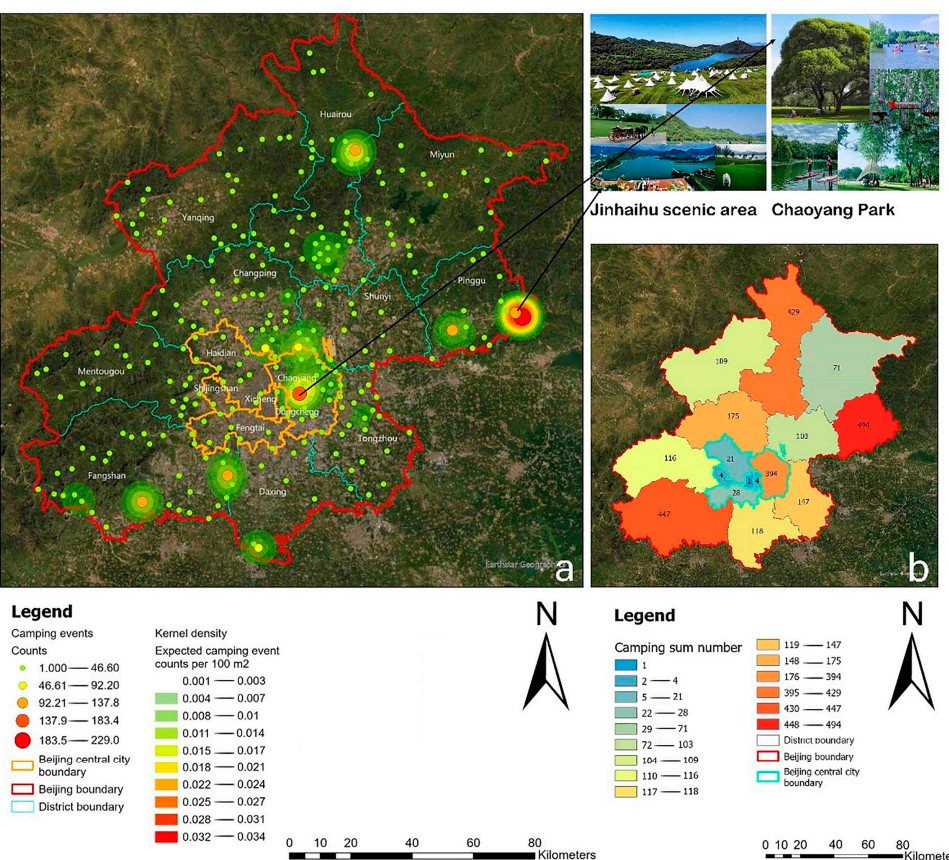

**Figure 2.** (**a**) kernel density heat maps of camping notes and (**b**) the spatial distribution of camping notes in Beijing.

## 3.2. Distribution of Hot and Cold Points of RES Perceived by the Public

Figure 3 shows the spatial distribution of hot/cold points classified based on the number of "likes" received from the public. The hot points are statistically significant clusters of sites with greater RES values (H-H value), whereas the cold points are statistically significant clusters of locations with lower RES values (L-L value). Using Anselin Local Moran's I in ArcGIS, this analysis identified 98 hot points and 427 cold points. The hot points were observed to be dispersed throughout Beijing (Figure 3a), indicating either suburban or central urban regions. In the suburban area, the hot points had an inward-to-outward spreading pattern: high RES values of hot spots were associated with scenic areas in beautiful and unique mountainous and water landscapes to the north of Beijing, including Jihaihu scenic area and Yudushan scenic areas (referred to Table 2). In the central urban area, hot points were also identified. According to POI (Table 2), these hots points were featured by forest parks and wetland parks, such as the Dayunhe Forest Park and Qinglonghu Wetland Park. Among these, Yudushan scenic region and Jinghaihu scenic area, with nearby camping sites, have the highest number of "likes" with 6661 and 5079, respectively (Table 2). Interestingly, both gorgeous regions accommodate different outdoor recreational activities. For instance, visitors are not only permitted to pitch tents and

hammocks along the banks of large lakes for long-term stays; additionally, the recreational public facilities support picnicking, boating, carriage, horse riding, flying kites, walking dogs, and playing frisbee; and grassland concert events were also constructed accessible to the general public.

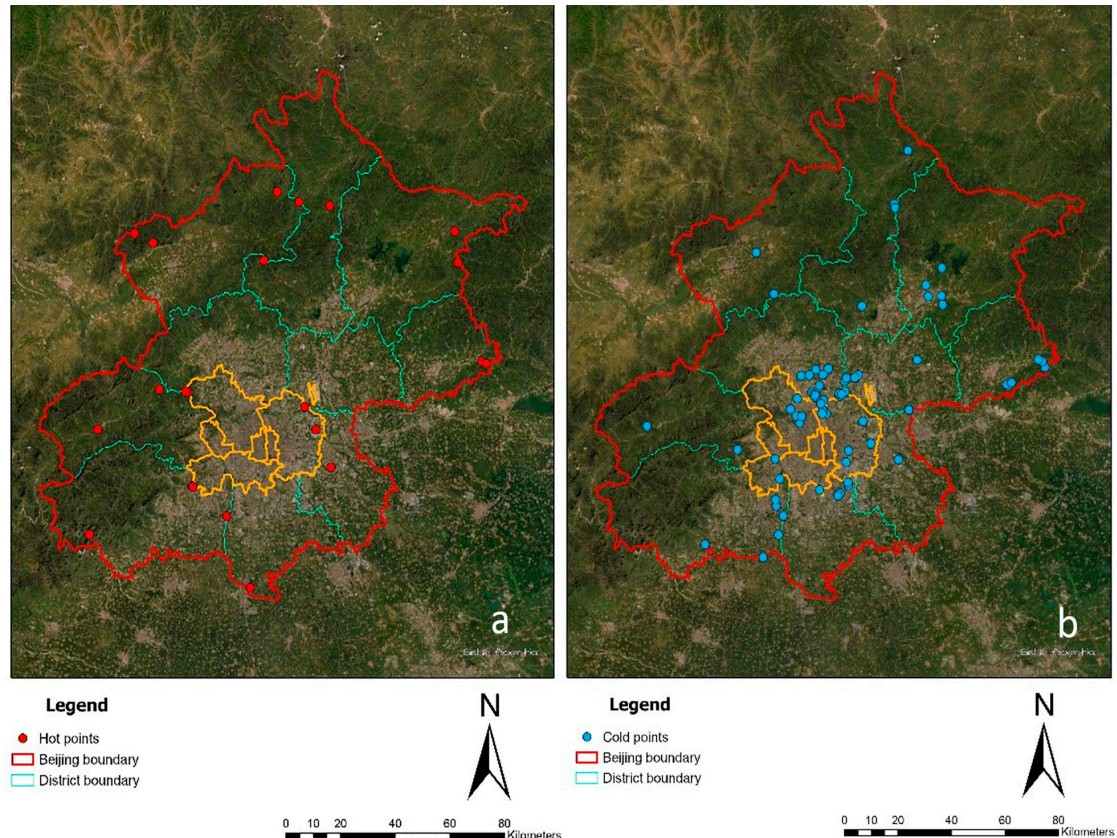

**Figure 3.** (**a**) Hot points per the RESs provided by camping notes. (**b**) Cold points per the RESs provided by camping notes.

**Table 1.** Descriptive statistics of the six highest-density areas.

| No. | Locations | Districts | Landscapes | Total Camping Notes | Percentage | Kernel Density (Mean) |
|---|---|---|---|---|---|---|
| 1 | Jinghaihu scenic area; Weilan Valley camping park | Pinggu | Lake, mountain, forests, lawn | 356 | 12.28% | 0.0162 |
| 2 | Chaoyang Park | Chaoyang (central urban area) | Urban park, lake, forest, lawn, European buildings | 504 | 16.69% | 0.0084 |
| 3 | Baihewan scenic area | Huairou | Mountain, forests, riverside, bare land | 183 | 6.15% | 0.0093 |
| 4 | Daxiangludao resort | Pinggu | Lawn, woods, lake, | 107 | 3.60% | 0.0070 |
| 5 | Tiankai Reservoir, Tiankai Farm | Fanshan | Mountain, bare land, cultivated land, pond | 158 | 5.32% | 0.0081 |
| 6 | Shuangying camping park | Fanshan | Grassland, bare land | 137 | 4.61% | 0.0083 |

**Table 2.** Descriptive statistics of the top three locations with the highest RES value as perceived by the public.

| No. | Poi Name | Landscape Features | Recreation during Camping | *n* = Like |
|---|---|---|---|---|
| 1 | Yudushan scenic area and local Yudu camping park | Mountain, forests, lake, grassland, lawn, pool, waterfall | Tenting, picnicking, boating, hiking, dog walking, barbecues, sawanobori | 6661 |
| 2 | Jinhaihu scenic area and local Weilan Valley camping park | Lake, mountain, forests, lawn | Tenting, picnicking, boating, carriage, horse riding, flying kites, walking dogs, and playing Frisbee, and grassland concert events | 5079 |
| 3 | Darehuangye | Grassland, bare land, mountain | Tenting, picnicking, campfire, barbecues, motorbikes, concerts, and market events | 2726 |

In contrast, the cold points were mainly found in the central urban areas, particularly in the northern Haidian and Chaoyang districts (Figure 3b). Intriguingly, we discovered that the cold points near the north Haidian, Chaoyang, and Changping districts (Table 3) indicated the presence of numerous local forest parks and country parks, including Olympic forest park, Wenyuhe park and Dongxiaokou country park. Numerous visitors have complained about local restrictions in these campgrounds, including prohibitions on dog walking, boating, horseback riding, and swimming, which weakens their recreational support for the public. Additionally, cold points were spotted in the suburban area. Those points mostly referred to scenic areas and various natural landscapes developed as camping parks by private companies or local village committees (Table 3). According to our survey of campers' submitted comments, this is owing to their differing degrees of facilities and design in relation to the area landscape, as well as their inadequate maintenance. For instance, the camping parks and villages around Jinhai town and Wensuhe tend to have inferior architecture, recreational facilities, and maintenance, such as a lack of water sources, restrooms, parking, and waste management.

**Table 3.** Descriptive statistics of the cold points of the top three high RES value locations as perceived by the public.

| No. | Poi Name | Landscape Features | Recreation during Camping | *n* = Like |
|---|---|---|---|---|
| 1 | Huangye No.7 camping park, Jinhaihu town | Mountain, lake, grassland | Tenting, picnicking, hiking, dog walking | 242 |
| 2 | Xiaoqinghe | Woods, wetland, forests, riverside, bare land | Tenting, picnicking, BBQ | 121 |
| 3 | Wenyuhe | Woods, riverside, bare land, sand land | Tenting, picnicking, dog walking | 117 |

*3.3. Relationship between Hot/Cold Spots, Land Cover Features, and Visitor Genders*

Correspondence analysis enabled us to evaluate the links between the six categories of land cover features, the sites where people perceive high-high RES values (hot points) and low-low RES values (cold points), and the gender of visitors. Figure 4 shows that component 0 explains 17.7% of the total variability along the *x*-axis, which proved to

be the most important (major) contribution in this correspondence analysis. In contrast, component 1 explains 15.90% of the total variability along the *y*-axis, which comes in the second place. The outcome of Figure 4 indicates that the perceived hot points locations were significantly related to the grassland and forest. In contrast, perceived cold point locations tended to be associated with wetlands and waterbodies. In addition, female visitors tend to favor grassland during all of these activities, which results in high or low RES values as a result of their camping activity. Instead, males are significantly more interested in built-up areas and bare land.

**Figure 4.** Correspondence analysis of hot and cold points relating to the six land cover types and all districts in Beijing.

## 4. Discussion

The study evaluated the public's perception of the RES given by urban green areas based on camping behaviors shown by social media data from the megacity Beijing. Here, we discuss our key findings: (i) the spatial pattern of camping activity, (ii) the public's perception of RES values, and (iii) the elements that influence RES values. In addition, we address the advantages and disadvantages of employing social media datasets for RES assessments in comparison to conventional approaches.

### 4.1. The Spatial Pattern of Camping Behaviors

Our spatial pattern of camping behaviors revealed that, excluding the urban area, a significantly greater number of camping behaviors were observed in the suburban area, which is comprised of picturesque locations and attractive natural landscapes. This is attributed by the author to the desire for privacy and tranquility, which inspires many people to go camping and explore nature, and as a result, empty and calm landscapes were favored globally [32]. Similarly, according to previous studies [33], a disproportionate number of social media posts are uploaded in landscapes with high scenic quality, such as national parks and river corridors. In addition, our study also addressed that the high density of camping behaviors is not only featured in suburban areas but also in the urban park in the central urban area. Similar to the findings of earlier studies, distance and transit variables could be primary factors that affect public activities such as visiting parks for

natural recreation [34]. The proximity and accessibility of these urban parks in the central urban area also make them appealing camping locations for visitors.

*4.2. The Spatial Pattern of Public Perceived RES Value*

In accordance with the above finding, our hot/cold point analysis results revealed that the hot points concentrate around suburban areas. In addition, we discovered that other than the landscape elements provided, the local management policy for supporting camping and outdoor recreation can be another essential factor that determines high RES values. Similar findings were addressed by previous studies [32,34]. For example, the notes posted about the Jinhaihu Weilan Valley camping park (a private recreational park operated by a private company) demonstrated a high level of appreciation for available recreational activities such as camping, picnicking, boating, carriage and horseback rides, kite-flying, walking dogs, playing Frisbee, fishing, swimming, and kayaking in the lake, as well as concerts and an outdoor market that utilize the local landscape. In contrast, cold points were clustered in the central urban area featured with urban parks, forest parks, and country parks. Further, some cold points were discovered in an urban location with a high density of camping behavior. This could be attributed to recreation tensions between public recreational desires and the RES offered by urban parks. In contrast to previous conflicts between campers and local inhabitants [35], we believe that the conflicts between managers and campers in these places stem from the design and management philosophy of urban parks at the highest level. Apparently, the local urban park administration appears to have limited visitors' access to varied leisure possibilities, which is not the ideal option for attaining the maximum CES benefit for the public. For example, the camping notes located in the Olympic Park noted, "We came here for camping, and learned that tents, hammocks and picnic mats are not allowed in the area". "We had to camp secretly while the security staff were not patrolling". Furthermore, practically all urban park settings in Beijing have a pet prohibition, which dramatically reduces the perceived RES value. These leisure activities were labeled as "uncivilized" in the 2002 Beijing Park Regulations. Previous CES studies, on the other hand, found that dog walking and picnics are major recreation ecosystem service indicators that influence CES assessment patterns [36]. Picnic and camping areas are also seen as important features in the cultural landscape and green space management by several scholars [32,37,38]. In order to improve RES values for the majority of urban parks, it is therefore beneficial to reconsider urban park management regulations in order to suit the ever-changing aspirations of Beijing residents as the post-COVID-19 era continues.

*4.3. Impact Factors towards Public-Perceived RES Value*

According to the results of the correspondence analysis, the occurrence of hot points was highly associated with grasslands and woods. Several existing CES evaluation studies also demonstrate similar results. For instance, scholars [37,39,40] documented the significant contribution of grasslands and forestry lands towards RES value. In addition, previous research [1,8,40] identified water bodies as an important land cover that correlates to better recreation values and enhanced leisure possibilities. However, our results indicated that cold points were associated with wetlands and waterbodies, as camping notes indicated that visitors did not take advantage of the recreational opportunities given by surrounding waterbodies. It could be explained by the many limits on free enjoyment in public water bodies, in addition to the environmental protection of water quality in local water bodies. Since 2020, for instance, the Water Bureau has restricted not only swimming and fishing within water conservation zones but also gatherings, open-air barbecues, and picnicking around public waterbodies, such as rivers and lakes throughout the entire city [41]. Specialized prohibitions were placed on "wild swimming," which is criticized as "uncivilized behaviors" in the event of swimming in urban rivers and park lakes, as well as splashing water. This considerably restricts people's access to waterbodies; hence, the recreational value of public water places is significantly diminished. Conflicts between public leisure

needs and public waterbody management have been the subject of a growing number of recent discussions and news reports [42–45]. In contrast to Beijing, many other cities, such as Copenhagen and Berlin, have permitted the establishment of dedicated outdoor water recreation areas, particularly for wild swimming, on a number of the city's natural waterbodies, such as urban rivers, canals, lakes, and harbors [46–48]. Numerous studies also highlight the multiple recreational benefits of urban water bodies [49,50], as well as the fact that water recreation activities such as swimming, fishing, and boating (including paddling and canoeing) can be used as an indicator for assessing the value and benefits derived from the public waterbodies' environment [51]. In order to increase the RES values in public waterbodies, we believe that the public sector should evaluate the reasonable requests of the public to swim and play in the water. In the future, a proper management system should be established to balance public expectations, public safety, water resource protection, and public safety.

Meanwhile, our findings suggested that gender significantly affects landscape preference, with female visitors preferring grassland and male visitors preferring bare ground and urban areas for camping enjoyment. Similar findings were reported in prior studies [52–55]. Particularly, the study regarding Phoenix revealed that women were much more averse to dry settings devoid of lush vegetation [55]. In addition, researchers have proposed gender variations in recreational use and views of nature [56–58]. "The lawn and grassland here is very good to be portrayed, with my newly purchased picnic cloth and picnic basket, very English style landscape oil painting scene feeling", stated a female visitor who posted her camping notes in a grassland area. In addition, the phrase "nice to be portrayed" appears frequently in the camping notes of female visitors. Similar to the study in Georgia [56], women emphasize photography substantially more than recreational activities such as hiking and bird viewing, which are more frequently addressed by men. Females are more likely to seek out campsites with luxuries, whereas males are more likely to focus on the activities the campsite has to offer and do duties peculiar to camping [34,59]. This could explain our observation that male visitors are more interested in barren land. "No campfire, no camping. Without vegetation, this gravel-filled location is suitable for campfires, and we also played mountain biking and motorbikes here", according to a male camper who enjoyed a unique camping experience on bare terrain.

In addition, we found that people camped in urban areas near pocket green spaces, their own residential districts, and universities. These sites provided convenient opportunities for people to experience nature in an urban context, although the camping behaviors of visitors may cause potential issues. Current various debates and news articles in China mention these conflicts regarding whether camping in open green spaces in residential districts and communities is detrimental to residents' public health and whether school administrators should prohibit students from camping on university grounds during school closures [60–62]. Previous research found that concerns about conflicts with others, personal safety, and harassment significantly reduced female participation in outdoor recreation, leading to the conclusion that women do not have as strong an interest in camping in built-up areas as men, despite the fact that it provided convenient green spaces for staying in nature, whereas males were not as concerned [63–65].

### 4.4. Advantages, Limitations, and Future Potentials for CES Evaluation Based on Mapping Geo-Tagged Camping Notes from Social Media Data

Our study demonstrated a crowdsourcing data approach to evaluating RES provided by urban green space throughout camping behavior. Such practices provide planners with a new resource for quantifying and comprehending the RES values given by existing green spaces. This enables decision-makers to maximize their policy-making actions, which better satisfies the public's rapidly changing expectations for outdoor leisure and enhances the well-being of individuals in densely populated megacities. In contrast to prior spatial assessments of RES, which were typically based on expert viewpoints or used land cover as a proxy [66,67]—our study spatially examined RES using public perspectives.

At this point, the public data collection activities allow citizens' preferences and concerns to be incorporated into the development and management of green space; this is known as public engagement. Such practices are regarded as quite valuable in China, where the mainstream planning process is viewed as a top-down technological engineering process based solely on the views and knowledge of decision-makers and experts, as opposed to a social process of cooperation with various stakeholders [68]. For many public participation studies conducted to date, the time-consuming and expensive nature of these surveys and interviews is viewed as a barrier that reduces planners' motives to involve public participation [69]. Currently, social media data is widely accessible and updated every minute as new data becomes available, which demonstrates considerable benefits. Consequently, a rising number of academics employ social media gathered from platforms such as Panoramio and Flickr [40] for tasks such as landscape value mapping, landscape aesthetics mapping, evaluations for establishing land use policy, and crowdsourcing data for CES assessment [40,70,71]. Nonetheless, the hunt for proxies that reliably identify RES for mapping the locations and attributes related to people's nature experiences and perceptions continues [3]. In this study, we used camping notes from the largest Chinese social media platform (LRB) for life sharing to explain the experiences of people who stay in the wilderness and acquire a variety of recreational activities from the local landscape. In addition, our study utilized social media data regarding camping notes and pertinent public perception for the evaluation of RES by giving a number of "likes". We consider not only the visitors' perceptions of RES value in their camping sites but also public perception and cognition of local RES affected by the geo-tagged photos and descriptions contents through the social media data, which could aid in the future exploration of the public's visiting preferences to various parks [32]. In contrast to conventional methods, we summarize the pros of utilizing social media camping data for public perception RES surveys as follows: (1) Social media data are readily accessible, enabling opportunistic sampling of visitors' expressed landscape values. (2) Social media data could reveal the public's perception of the landscape's value (e.g., cultural ecosystem services, recreational ecosystem services) over an extended period of time and on a broad scale than traditional interviews and surveys. (3) Social media data can capture large amounts of data in a brief amount of time to reflect public perceptions, making it a rapid and cost-effective method for identifying and mapping RES they gained in megacities.

However, the limits of social media data must also be acknowledged. For instance, geo-tagged photographs from social media sources may provide cellular signal strength uncertainties for electrical gadgets. According to Zielstra and Hochmair (2013), the accuracy of the geographical reference points for these devices can impact the quality of the data gathered [72]. The physical, perceptual, and evaluative aspects of landscapes are also difficult to measure from different user groups, and these are the limitations we must acknowledge. Although the Little Red Book (LRB) platform is currently the most popular social media platform in China, the majority of our collected data may only represent the perceptions of younger citizens regarding RESs and their natural experiences of green spaces, as the majority of LRB users (83.31%) are between the ages of 18 and 34 [73]. Citizens outside of this age range may not share as many remarks about RES locales on social media or other websites. The quantity of "likes" assigned to influencers (e.g., Key Opinion Leader (KOL) camping remarks) may further increase statistical uncertainty. These influencers may have a greater number of notes about recommended camping facilities than actual users, and their meticulously manipulated photographs and travel blogs rarely provide geo-tagged information about the camping place. Consequently, more stringent data cleaning is required to reduce the potential for bias in the outcomes resulting from correspondence. The statistical dependability and locational precision of crowdsourced camping notes must be examined in further research.

At the same time, even though we controlled the context of camping notes on the scenery and camping-related activities by deleting images of the user's personal features (such as the user's own portrait during camping events) in order to reduce the interference

caused by the person rather than the context. There are still limitations to the public impression possessed by the social media image, as pictures published in camping notes may tend to avoid unsightly areas by emphasizing the best features, and posts may misrepresent something as positive when it is not. Therefore, additional field research validation of the information reflected in social media data demands further investigation, and field surveys and interviews with the general public will supplement the social media data in future studies. In addition, the drawbacks of social media data used in the RES evaluation included the loss of individual-level user-specific information. Due to users' right to privacy, the social demographic profile, including education, age, degree of literacy, and social class and income, cannot be distinguished in social media data. Even though the user profiles of some social networks can describe the general characteristics of the social background of users, additional interviews or questionnaires are required to investigate how social context influences the recreational value that individuals derive from the natural environment in future research.

## 5. Conclusions

This study uses social media data to assess the RES value of urban green spaces in the meta-city of Beijing. The present spatial patterns of camping behavior after COVID-19 were explored using geo-tagged camping posts, and hot/cold points linked with high/low RES values were identified and distinguished across the entire city. Based on this, the public's preference for urban green space was analyzed and comprehended in terms of land cover and gender inequalities. Our primary findings are as follows:

- According to the spatial pattern of camping habits, both suburban and urban central areas have given certain camping support services to the public since the implementation of COVID-19 restrictions. However, much more clusters were detected in the suburbs than in the central districts, indicating that existing urban green spaces do not provide enough camping facilities.
- Hot points associated with high-RES values were mainly identified in suburban areas. Despite the high density of camping noted in the central urban area, the green spaces in this area provide relatively low RES value for the public. We believe this is due to the restricted public leisure activities supported by these urban parks as a result of their current management policies. Therefore, additional optimization is required.
- Differing from previous studies, the lower RES value was correlated with waterbodies in Beijing. Here, we think this is due to local recreation restorations of public water bodies. Higher RES were closely related to grassland and forests. In addition, female visitors tend to camp in grassland while males tend to bare land and built-up areas, which could be explained by the gender differences in landscape preferences, recreational use of nature, and concerns about conflicts.

The last two years have witnessed a significant increase in camping activities, which has altered the most popular form of outdoor recreation in green spaces. This appears to be a fast-mind transition that alters how people perceive available green spaces. Here, RES value assessment based on social media data can provide a more up-to-date and in-depth understanding of public demands and perceptions, thereby providing better inspiration for decision-makers to improve current green space management in order to adapt to such a dynamic need from the public. In suburban areas, our study highlighted both the high density of public camping and the high perceived recreational value, which reflects the potential for rural tourism in suburban areas. Accordingly, we propose that management policies for suburban green space should be integrated with local rural development and tourism policies, with a focus on the new model of "camping+", such as camping+ outdoor adventure activities in wild lands, camping+ trip photographing, camping+ cultural recreational activities (e.g., open-air concerts on the lawn or by the waterside), and other products with different characteristics for public preferences, in order to inject new momentum into rural development. In the central region, our research revealed the public camping demand and the limited recreational value obtained by campers. It indicated

conflicts between public new outdoor recreation and previous urban park management policies, such as the restriction on water sports recreation in blue spaces and the prohibition on picnicking in many urban parks. We propose that the management of urban green space reexamines the current one-size-fits-all park management regulations and that policies for open park use be formulated based on the landscape characteristics of different green spaces, such as forests, water bodies, grasslands, artificial ground, etc. For instance, open shared areas should be designated in areas such as park lawns and forested spaces, and supporting service facilities should be improved to better satisfy the needs of the public for outdoor activities such as camping and recreation close to nature, including the setting up of tents. In addition, the water surface's suitability for recreation could be evaluated, and water-opening zones and management policies should be developed to satisfy the public's demand for water-friendly outdoor sports and fitness activities. However, additional research is still needed to comprehend the long-term effects of these camping patterns on urban green areas since 2020. Beyond the RES value, these tendencies may have an impact on the other CES, such as aesthetics, natural education, and a sense of belonging, which may result in further difficulties. The following aspects are therefore proposed for future research:

(1) As green areas provide varied perceptions of CES, the conflict between campers and stakeholders could be investigated. (2) With different temporal scales, variations in RES value before and after camping would be interesting to investigate and compare in the future; (3) With a diverse range of group users (e.g., ages, education level, and agenda) in social media datasets, the different perceptions of different groups towards green space would be interesting to investigate and compare in the future.

**Author Contributions:** Conceptualization, H.X. and M.M.; methodology, H.X.; software, G.Z.; validation, Y.L.; formal analysis, G.Z. and M.M.; investigation, G.Z.; resources, H.X.; data curation, M.M.; writing—original draft preparation, H.X.; writing—review and editing, G.Z.; visualization, H.X.; supervision, H.X.; project administration, H.X.; funding acquisition, M.M. All authors have read and agreed to the published version of the manuscript.

**Funding:** This research was funded by [Research Enhancement Project for Young Scholars of BUCEA] grant number [x21044] and [National Natural Science Foundation] grant number [52008171].

**Data Availability Statement:** Not applicable.

**Conflicts of Interest:** The authors declare no conflict of interest.

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
