# Peer review of "Using Social Media Camping Data for Evaluating, Quantifying, and Understanding Recreational Ecosystem Services in Post-COVID-19 Megacities: A Case Study from Beijing"

_forests, doi:10.3390/f14061151_

Round 1

Reviewer 1 Report

This paper illustrates the use of social media to study the Recreational Ecosystem Services. The work examines the green areas of the Beijing mega city.

Although this work certainly has the merit of being able to systematise the recreational and camping areas of a mega-city like Beijing (something that would be very difficult with traditional questionnaire or interview techniques used in the social sciences), it has various limitations, some of which have been pointed out by the authors others are those I have listed in the Discussion section and which should be added to and discussed.

Introduction and methods

From a structural point of view, the article has no problems, it is well structured, the introduction is well done, and the methodology is well explained, may be could be useful give more details in the following points:

a) better explain the difference between hot and cold spots in the methodology; it is true that the authors say that it is due to the significance calculated by the software: but what degree of significance? at what level is it decided that a spot is hot or cold? Based on what values?

b) line 118: it would be interesting to have a more detailed description of the city: how many inhabitants does the historic centre have and how many in the suburban areas; have new parks been set up in the post Covid period? what are the policies for the use of these areas, i.e. is it possible to camp everywhere? 

c) line 226: it says 'we question the users who posted their campaign notes....Did you conduct interviews? How many? Where is this data?

Results

The results of the Principal component analysis could be better explained, there are no tables or related data to better understand the interpretation given by the authors.

Discussion

Limitations of this work.

Among the major limitations of the work not mentioned by the authors there is the fact that the use of 'likes' as a measure of liking is very arbitrary as likes are also placed out of sympathy for the person posting and not necessarily because they like that place or what is being said. This happens even if the person is not an influencer. Likes do not give a real idea of liking because they are mediated by the impression that passes through the social media and that can be distorted because published photos often tend to avoid ugly places by enhancing the best features. Moreover, posts always tend to pass off as positive something that is not, thus influencing likes.

Other limitations of the work include the fact that the degree of literacy and/or social class of the participants cannot be distinguished, the socio-demographic profile of the users is missing, which can only be done with interviews or personal questionnaires.

References

The list of references seems to be made automatically and not checked afterwards.  All the references to scientific journals for example are missing, there is only the number and pages but not the name; the websites from where some references were taken are missing, the date of access is missing.

Author Response

REVIEWER1

This paper illustrates the use of social media to study the Recreational Ecosystem Services. The work examines the green areas of the Beijing mega city.

Although this work certainly has the merit of being able to systematize the recreational and camping areas of a mega-city like Beijing (something that would be very difficult with traditional questionnaire or interview techniques used in the social sciences), it has various limitations, some of which have been pointed out by the authors others are those I have listed in the Discussion section and which should be added to and discussed.

Introduction and methods

From a structural point of view, the article has no problems, it is well structured, the introduction is well done, and the methodology is well explained, may be could be useful give more details in the following points:

  1. better explain the difference between hot and cold spots in the methodology; it is true that the authors say that it is due to the significance calculated by the software: but what degree of significance? at what level is it decided that a spot is hot or cold? Based on what values?

Many thanks for your comment. we added more explanation regarding the differences of hot and cold spots in the method. Please see the new explanation in 2.3.2 line 262-292.

  1. line 118: it would be interesting to have a more detailed description of the city: how many inhabitants does the historic centre have and how many in the suburban areas; have new parks been set up in the post Covid period? what are the policies for the use of these areas, i.e. is it possible to camp everywhere? 

Thanks for the suggestions, we added the line 144-155 and new Figure.1b (in 2.1) to elaborate more the population of inhabitants in Beijing. We also added more contents regarding the new parks in Beijing in the post Covid and the policies of camping management in urban parks in Beijing. (Please see lines184-203).

  1. line 226: it says 'we question the users who posted their campaign notes....Did you conduct interviews? How many? Where is this data?

 Thanks for the comments, we added more detailed information about the interview part in the Section 2.3.2. Please see lines 293-300.

Results

The results of the Principal component analysis could be better explained, there are no tables or related data to better understand the interpretation given by the authors.

Thanks for the comments, we think there might be some misunderstanding hereof. In fact, we used correspondence analysis instead of principal component analysis. Yet, to avoid potential confusion to readers, we added more explanation of correspondence analysis (including cases used in previous studies) in the methods lines 303-324, and we also added more explanation of results with related data at lines413-422.

Discussion

Limitations of this work.

Among the major limitations of the work not mentioned by the authors there is the fact that the use of 'likes' as a measure of liking is very arbitrary as likes are also placed out of sympathy for the person posting and not necessarily because they like that place or what is being said. This happens even if the person is not an influencer. Likes do not give a real idea of liking because they are mediated by the impression that passes through the social media and that can be distorted because published photos often tend to avoid ugly places by enhancing the best features. Moreover, posts always tend to pass off as positive something that is not, thus influencing likes. Other limitations of the work include the fact that the degree of literacy and/or social class of the participants cannot be distinguished, the socio-demographic profile of the users is missing, which can only be done with interviews or personal questionnaires.

Many thanks for your great comments and suggestions. We added some paragraph to explain how we tried to reduce the influence in the data processing part (please line see231-236). But we have to acknowledge that these relevant limitations cannot be avoided and thus more future research are still required to improve these potentials. (Please see line 614-630)

References

The list of references seems to be made automatically and not checked afterwards.  All the references to scientific journals for example are missing, there is only the number and pages but not the name; the websites from where some references were taken are missing, the date of access is missing.

Sorry about that. We have already manually reviewed all the citations and inserted the names of all the journals and website addresses (with access date). (Please see the references)

Reviewer 2 Report

Title - OK

Abstract - OK

Introduction

Line 55 and 56, change Brown 55 et al. (2014) by [7]

Line 58, change Xu et al. (2020) by [8]

Lines 67 and 73, Van Berkel et al. and Schirpke et al, need to be numbered.

Material and Methods

Line 201, change cell size of 100 m by cell size of 100 m2; or cell size of 10 m

Line 202, change count in 100 m2 cells by count in 10,000 m2 cells.

Lines 225, 226 and 227 “Furthermore, we questioned the users who posted their camping notes in these hot and cold points locales for their particular camping experiences there”. Insert a brief text about the methodology used

Results

In line 323 change Fig. 6 by Fig. 4

Discussion - OK

Conclusion - OK

Author Response

Reviewer2 Title - OK

Abstract - OK

Introduction

Line 55 and 56, change Brown 55 et al. (2014) by [7]

Line 58, change Xu et al. (2020) by [8]

Lines 67 and 73, Van Berkel et al. and Schirpke et al, need to be numbered.

Thanks for the suggestions, we deleted all the authors name and use “some authors and [reference number] “ to replace them for what they have done in their studies and renumbered them as your suggestions. (Please see line 57,60, and 69-70).

Material and Methods

Line 201, change cell size of 100 m by cell size of 100 m2; or cell size of 10 m

Line 202, change count in 100 m2 cells by count in 10,000 m2 cells.

Thanks for this suggestions and sorry for these faults, we already modified that. Please see line 259-260.

Lines 225, 226 and 227 “Furthermore, we questioned the users who posted their camping notes in these hot and cold points locales for their particular camping experiences there”. Insert a brief text about the methodology used

Thanks for this suggestions, we added an explanation text about the methodology we used, please see line 294-301.

Results

In line 323 change Fig. 6 by Fig. 4

Thanks for the reminding, we already modified it. Please see line 413.

Discussion – OK

Reviewer 3 Report

The manuscript has an interesting theme on RES with data from social media camping data.

1. In the Instruction, the authors presented three study questions. Please provide more specific justification for the questions, relating to the contribution of the study to recreational policy and management.

2. Please provide the pros and cons of the survey methods using social media camping data.

3. In Conclusion, provide more implications in recreation policy and management relating to the results the study found.

Author Response

The manuscript has an interesting theme on RES with data from social media camping data.

  1. In the Instruction, the authors presented three study questions. Please provide more specific justification for the questions, relating to the contribution of the study to recreational policy and management.

Many thanks for the suggestions, we then added three specific justifications for the three questions relating to the contributions of their answers to recreational management policy in the introduction part. Please see line105-128.

  1. Please provide the pros and cons of the survey methods using social media camping data.

 Thanks for the suggestions, we added more paragraphs about the pros of using social media camping data for our RES assessment, please see line 586-594. And even though we addressed some cons in our limitation parts in 4.4, we added more new cons there, please see lines618-630.

  1. In Conclusion, provide more implications in recreation policy and management relating to the results the study found.

Thanks for the suggestions, we've added an additional paragraph discussing the implications of our findings for recreation policy and green space management in both suburban and urban areas. Please see lines 661-684.
